# Growth Media Efficacy in Biochemical Methane Potential Assays

**Giles Chickering * and Thabet Tolaymat**

Office of Research & Development, US Environmental Protection Agency, 26 Martin Luther King Dr W, Cincinnati, OH 45268, USA
* Correspondence: chickering.giles@epa.gov; Tel.: +1-513-569-7256

**Abstract:** The Biochemical Methane Potential (BMP) assay is a vital tool for quantifying the amount of methane that specific biodegradable materials can generate in landfills and similar anaerobic environments. Applications of the protocol are extensive and while simple in design, the BMP assay can use anaerobic seed from many different types of sources to determine the methane potential from most biodegradable substrates. Many researchers use differing protocols for this assay, both including and excluding the use of synthetic growth medias, intended to provide vital nutrients and trace elements that facilitate methanogenesis and leave the substrate being tested as the only limiting factor in methane generation potential. The variety of previous approaches inspired this effort to determine the efficacy of adding synthetic growth media to BMP assays. The presented findings suggest the use of M-1 synthetic growth media, defined in this study, at a volumetric ratio of 10% active sludge: 90% M-1 media yielded optimal results in terms of gas yield and reduced variability.

**Keywords:** biochemical methane potential assay; landfill gas; anaerobic digestion

## 1. Introduction

Variations of the biochemical methane potential (BMP) assay have been used to quantify methane yielded from biodegradable materials under anaerobic conditions since at least the late 1970s [1]. The method was later adapted for specific applications in agricultural and municipal solid waste (MSW) methane potential research [2,3]. Modifications to the BMP protocol over time include ASTM International publishing a standard method E1196-92 before later removing the method from circulation [4]. A subsequent ASTM method, D5210-92, for determining the anaerobic biodegradation of plastic materials in the presence of municipal sewage sludge was heavily based on the 1992 method and subsequently withdrawn in 2016 [5]. Similar methods include guidelines published by the US Environmental Protection Agency (EPA) in 2008 for BMP assays of organic chemicals [6]. Other methods include the VDI 4360 and similar official methods [7].

Despite the applicability of BMP assays to municipal solid waste research, one universally accepted standard method is not currently prevalent. Several researchers have reviewed past works in efforts to assemble updated methods and standardize the practice [8–11]. Most published research available includes methods that generally follow the original protocol of Owen et al. [1] with modifications of sample or vessel size, organic loading rate, anaerobic sludge source, and/or the addition of growth media to facilitate microbial activity [6,8,12]. Some of these factors have been reviewed and compared as more work has been published, specifically focusing on pH, moisture content, temperature, redox conditions, and other conditions that can impact methane yield in a closed system [13].

One factor in the methods that consistently receives less attention in efforts to improve these assays is nutrient addition via growth media. The goal of this work was to improve the scientific community's methods of assessing the methane potential of organic materials by determining the efficacy of synthetic growth media in BMP assay protocols. To accomplish this task, BMP assays were carried out under mesophilic (37 °C) conditions in anaerobic

serum bottles using growth medias based on published research and standard methods. The basis of these methods includes adding a mixture of nutrients, vitamins, and trace elements to prevent methanogenesis from being limited by nutrient availability, leaving the substrate being studied as the limiting factor in the assay. In other methods for BMPs and similar assays, some researchers have forgone the use of synthetic growth medias. This project sought to determine if adding synthetic growth media to a BMP assay is a necessary and valuable step.

## 2. Experimental Design

### 2.1. Experiment 1: Sludge Dilutions in Water

An initial experiment was performed using only anaerobic digester seed sludge and no additional growth media or nutrients. This tested the abilities of recently collected sludge to break down substrates and generate similar ultimate methane potentials from otherwise uniform conditions. The only modification to each trial, completed in triplicate bottles, was the concentration of sludge in deionized (DI) $H_2O$, as described in Table 1. Mesophilic sludge, sourced from the City of Fairfield, OH WWTP and maintained in the laboratory digester, was added to serum bottles with SuperQ ultrafiltered water with powdered cellulose as the substrate. Blank bottles contained no cellulose. Bottles were sparged with nitrogen prior to capping to form anaerobic conditions. All bottles were stored statically in a 37 °C incubator.

**Table 1.** Experimental conditions. Each row represents conditions in each triplicate set of bottles measured in this experiment. The % sludge, water, and media are all volume percentages. Cellulose and Xylose were lab-grade, powdered form from chemical suppliers. Food waste was collected, blended, freeze-dried, and ground in a blender before use as a substrate.

| Experiment | Condition | Substrate | % Sludge Inoculant | % Water (Type) | %Media (Type) |
|---|---|---|---|---|---|
| Experiment 1: Sludge Dilutions in Water | 1 | Cellulose | 100 | 0 (DI) | - |
| | 2 | Cellulose | 50 | 50 (DI) | - |
| | 3 | Cellulose | 10 | 90 (DI) | - |
| | 4 | Cellulose | 5 | 95 (DI) | - |
| Experiment 2: Sludge Dilutions in Media | 1 | Cellulose | 100 | - | 0 (M-1) |
| | 2 | Cellulose | 50 | - | 50 (M-1) |
| | 3 | Cellulose | 10 | - | 90 (M-1) |
| | 4 | Cellulose | 5 | - | 95 (M-1) |
| Experiment 3: Various Growth Medias with Various Substrates | 1 | Cellulose | 10 | - | 90 (M-1) |
| | 2 | Cellulose | 10 | - | 90 (M+V) |
| | 3 | Cellulose | 10 | 90 (SuperQ) | - |
| | 4 | Cellulose | 10 | 90 (Tap) | - |
| | 5 | Xylose | 10 | - | 90 (M-1) |
| | 6 | Xylose | 10 | - | 90 (M+V) |
| | 7 | Xylose | 10 | 90 (SuperQ) | - |
| | 8 | Xylose | 10 | 90 (Tap) | - |
| | 9 | Food Waste | 10 | - | 90 (M-1) |
| | 10 | Food Waste | 10 | - | 90 (M+V) |
| | 11 | Food Waste | 10 | 90 (SuperQ) | - |
| | 12 | Food Waste | 10 | 90 (Tap) | - |

### 2.2. Experiment 2: Sludge Dilutions in Media

Experiment 2 tested the same conditions as Experiment 1 with the substitution of M-1 growth media for DI water to demonstrate the impacts of adding growth media. The traditional formula paired 10% sludge by volume with 90% growth media. The goal of this experiment was to see if consistent and faster results could be obtained by modifying the sludge:growth media ratio.

### 2.3. Experiment 3: Various Growth Medias with Various Substrates

Experiment 3 was conducted to compare the BMP performance of growth media with and without an additional vitamin mixture ("M-1" and "M+V") to anaerobic sludge diluted to 10% with tap water and SuperQ ultrafiltered water. Based on the purpose of the growth media to provide an excess of nutrients, the hypothesis of this work was that the media with the most available constituents should produce the most successful methane assay, reaching its full potential in the shortest timeframe. The predicted order of most-to-least successful trials was M+V, M-1, Tap, and finally, SuperQ, based on the assumed availability of vital nutrients, trace elements, and buffering capacity. All media and water mixtures were autoclaved prior to loading with sludge and aliquoting to nitrogen-sparged serum bottles. All trials were completed in triplicate at mesophilic conditions.

In addition to testing the effects of growth medias relative to water on cellulose, additional experimentation was conducted to determine if the specified media in this study would produce different yields on a variety of substrates. Experiment 3 tested various medias and sludge dilutions in water on powdered lab-grade cellulose and xylose in addition to freeze-dried, ground food waste using the same BMP method applied in all other experiments.

### 3. Results and Discussion

The experiments in this study were intended to determine the efficacy of defined growth media in BMP assay protocols. Multiple permutations of growth medias, substrates, and sludge contents were assembled in BMP bottles to compare methane potentials under differing conditions.

### 3.1. Substrate and Sludge Analysis

The sludge used in this work was sourced from an active anaerobic digester at a wastewater treatment plant. This seed was maintained in a smaller digester in the laboratory and acclimated to the mesophilic conditions used for most of the experiments. In order to characterize both the sludge and substrates, these materials were analyzed for moisture content (MC) and volatile solids (VS) content using APHA 2540 methods for solids content [14]. Table 2 shows the findings of this analysis, completed with triplicate samples, which revealed the sludge to have a total solids content of about 1.4% and a volatile solids content of about 1.0%.

**Table 2.** Solids content analysis of substrates and sludge. Values are the averages of triplicate samples obtained via APHA 2540 [14].

| Substrate | Moisture Content | Total Solids | VS/TS | VS/Total Wet Sample Mass |
|---|---|---|---|---|
| Anaerobic Sludge | 98.6% | 1.4% | 72.1% | 1.0% |
| Cellulose | 2.4% | 97.5% | 97.6% | 95.3% |
| Xylose | 0.1% | 99.9% | 88.2% | 88.2% |
| Freeze-dried Food | 2.0% | 97.9% | 86.0% | 84.2% |

### 3.2. Experiment 1: Sludge Dilutions in Water

Experiment 1 investigated the ability of fresh sludge to complete BMP assays in varying dilutions with no growth media or additional nutrients added. Figure 1A shows

dilutions ranging from 100% sludge to 5% sludge, with the remaining volume in deionized water producing average methane yields between 348 and 383 mL $CH_4$/g cellulose volatile solid (VS). While the average ultimate methane potentials shown in Figure 1A are within 9% of each other, it is important to note that these values were normalized by subtracting the methane generation formed in the blanks, shown in Figure 1B. As an explicit example, the 100% sludge bottles produced an average of 132 mL of $CH_4$ before accounting for the average of 77 mL of $CH_4$ produced by the 100% sludge blank bottles. After subtracting the blanks to normalize and convert to STP conditions, the 100% sludge blanks produced a net average of 353 mL $CH_4$/g cellulose VS. The same calculations were performed with the 50% sludge conditions, which averaged 31 mL $CH_4$ in the blank bottles as well as the 10% and 5% blanks, averaging 3 mL $CH_4$ and 0 mL $CH_4$, respectively.

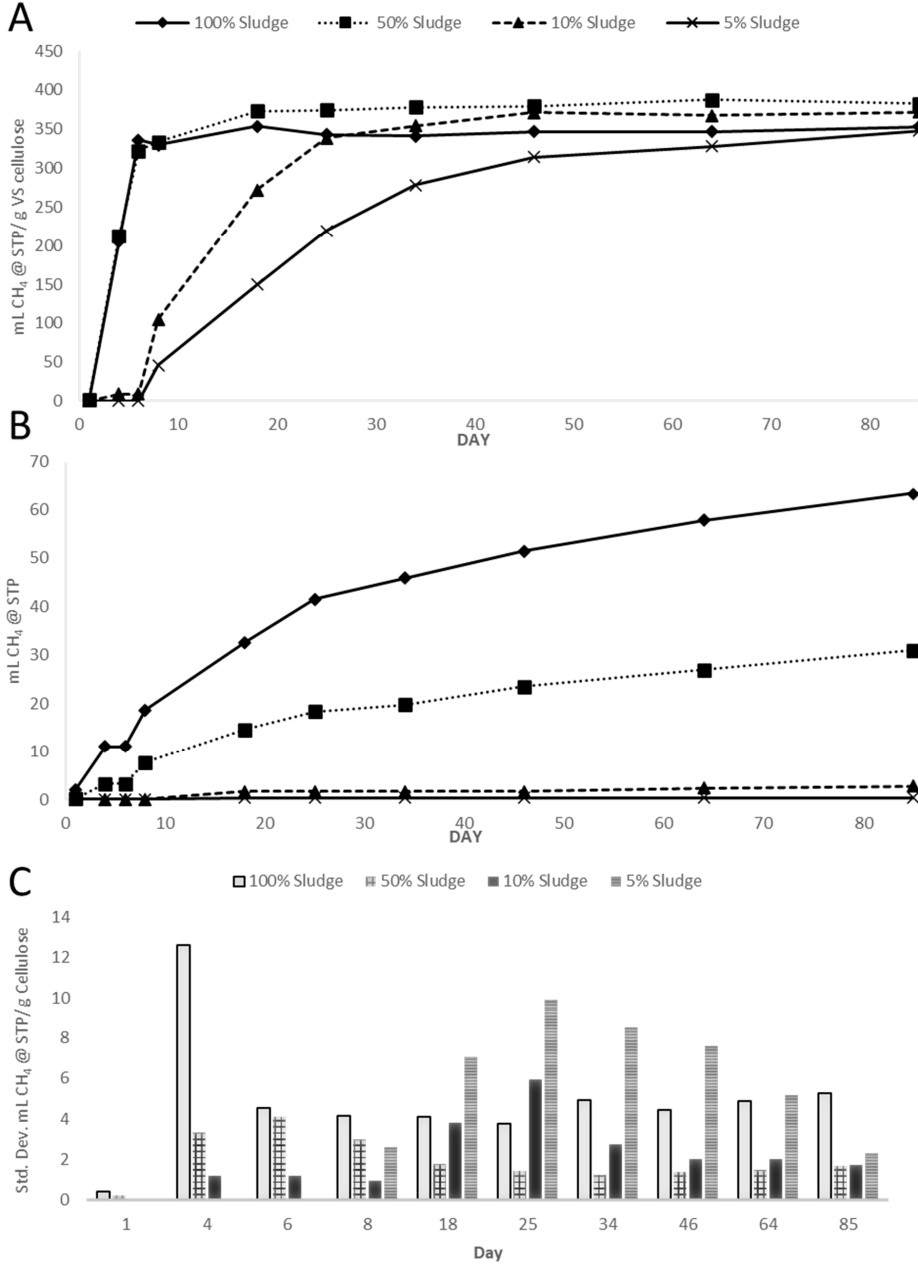

**Figure 1.** Sludge dilutions in DI water. (**A**) shows triplicate average methane yields of the experimental conditions, using cellulose as the substrate. (**B**) shows the blanks, which do not include a substrate. (**C**) shows standard deviations among triplicates represented in (**A**).

The curves of cumulative methane generation in Figure 1 show how 5% and 10% sludge samples showed a slower rate of generation, gradually catching up to the 50% and 100% samples over time. The lag phase of methanogenesis appeared to be more significant in the lower sludge concentration conditions. The 50% and 100% sludge blanks likely generated more methane due to the higher concentration of unmineralized organic matter in the sludge.

Another performance factor to consider was the range of variation among triplicate samples for each condition, shown as the standard deviation in Figure 1C. Early in the experiment, the 100% sludge BMPs showed the largest standard deviation, though this decreased over time with decreasing methane production rates. The 5% sludge bottles, conversely, took longer to approach their ultimate methane potential and showed increasing deviation among replicates as the bottles increased methane production. This limited number of critical methanogens in diluted sludge samples is expected to cause slower gas generation rates. The same conditions may also be responsible for variability as the success of methanogenesis is dependent on suites of microorganisms working in concert with each other and removing specific elements via sludge dilution is likely to cause variation. Standard deviations in the blank triplicates (not shown) were between 0 and 2 mL $CH_4$.

### 3.3. Experiment 2: Sludge Dilutions in Media

Experiment 2 replicated Experiment 1 with the substitution of M-1 growth media for DI water to demonstrate the impacts of adding growth media. The goal was to see if adding extra nutrients and constituents via growth media would normalize or compensate for sludge dilutions. The results in Figure 2 show nearly the same pattern as Figure 1, with the water experiments producing an average of 106% of the methane yields of the growth media experiments, all of which were within a similar spread; 348–383 mL $CH_4$/g cellulose VS in water, and 317–358 in M-1 media. The decrease in average yield in the M-1 bottles could be due to the experiment ending 9 days earlier than Experiment 1 due to unplanned circumstances. In both experiments, the 5% sludge bottle produced less methane than the other conditions, all of which were within 8% of the highest average yield.

As in Experiment 1, the blanks in Experiment 2 (Figure 2B) produced observable amounts of methane in the 100% sludge conditions, while the 50% sludge blanks in both experiments produced close to half as much methane as the 100% blanks. The 10% and 5% sludge blanks in Experiment 2 averaged 0 mL $CH_4$, avoiding the need to subtract background gas production while normalizing the results. Standard deviations of the triplicates for each condition in Figure 2C demonstrate decreasing variation for the 100% bottles over time, while the increasing gas production in the 5% conditions coincides with a trend in increasing variation. The 50% and 10% bottles averaged less variation among triplicates than the other conditions. Relative standard deviations in BMP assay replicates are known by the field to vary, from an optimized protocol developed by 30 participating laboratories of ±14% up to ±175% depending on the substrate and method (Hafner, Fruteau de Laclos, Koch, & Holliger, 2020; Holliger et al., 2016). The 10% sludge ratios used in these experiments all showed standard deviations among triplicates of 10% or less. The use of other sludge and substrates in the future could result in more variation; however, the consistency of these results relative to other studies is promising.

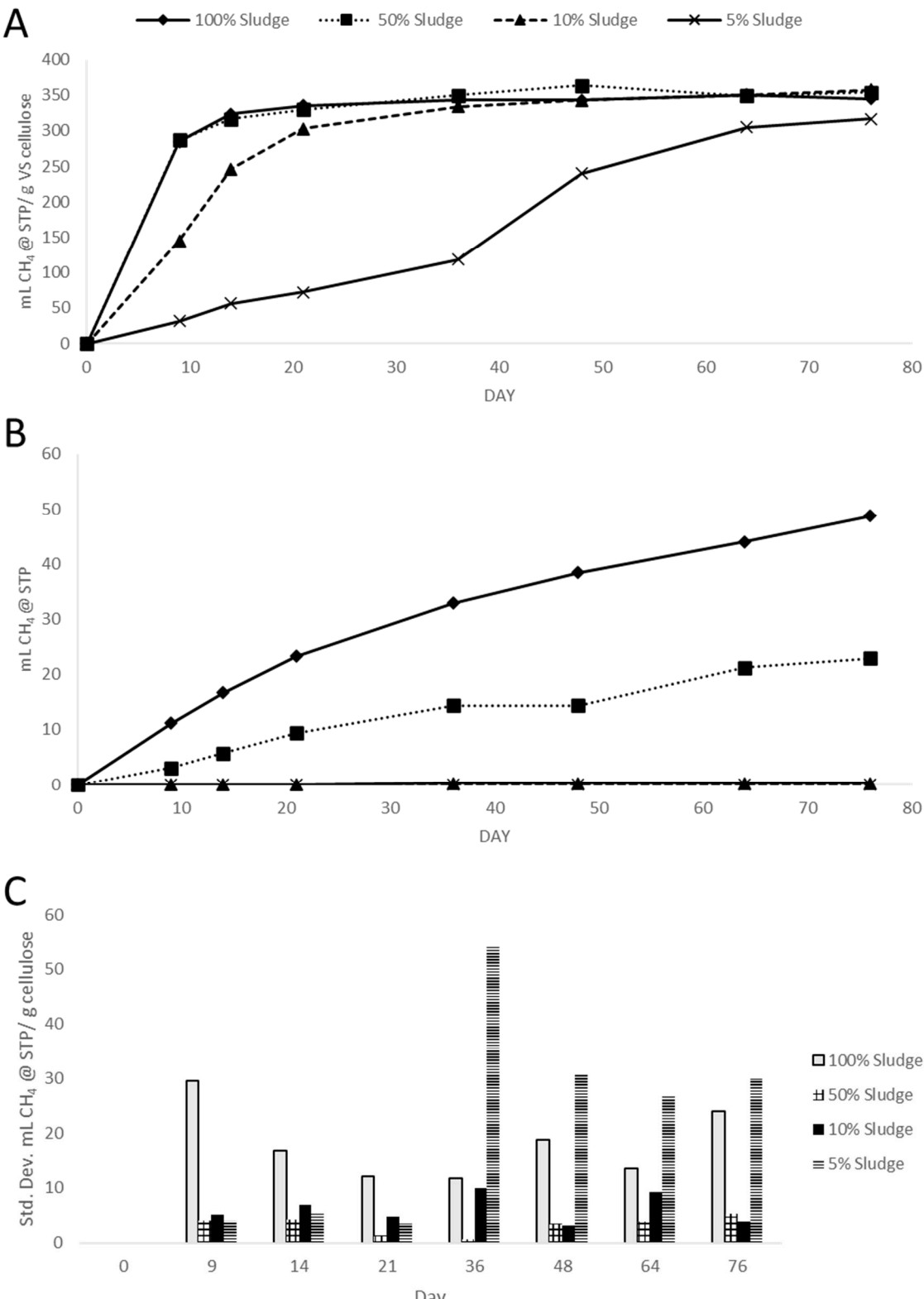

**Figure 2.** Sludge dilutions in M-1 Media. (**A**) shows triplicate average methane yields of the experimental conditions, using cellulose as the substrate. (**B**) shows the blanks, which do not include a substrate. (**C**) shows standard deviations among triplicates represented in (**A**).

### 3.4. Experiment 3: Various Growth Medias with Various Substrates

Reviewing the results in Figure 3 illustrated several important findings. Overall, M-1 media outperformed M+V media in every test for reasons that are not clear. M+V media has

all the same constituents as M-1, listed in Table 3, with only small traces (0.01 g/L at most) of several additional compounds. Unless an unknown error was made, the additional vitamin mixture in M+V should have either contributed nothing to the experiment or shown a slight improvement by supplying missing trace elements. It is unlikely, but evidently possible, that one of these added chemicals may have had a mild toxifying effect on the BMPs that were recurrent throughout this project. Potential reasons for the inhibition may have been mild toxicity caused by the overabundance of certain constituents or an unforeseen reaction caused by the comingling of many chemicals with the complex biological process of methanogenesis. Previous studied have also demonstrated that adding some mineral nutrient solutions can decrease BMP values by about 9% [15].

Another consistent finding was zero methane production by 10% sludge with 90% ultrafiltered (Millipore Super-Q) water in three of the experiments (food, xylose, and blanks) and significantly less methane production in the cellulose. Figure 3B illustrates how the SuperQ triplicates with cellulose as a substrate generated about 90% less yield than any other water or media. The suspected cause of this difference is the absence of trace minerals, salts, or any constituents that may buffer pH or lend nutrients to sludge when diluted to 10% in ultrafiltered water. The reason for having ultrafiltered water systems in laboratories is to remove these impurities, and the results suggest that the difference between tap water and SuperQ is substantial. Further, the difference between SuperQ and DI water conditions in Experiment 1 is also noteworthy, which produced 20 and 372 mL CH4/g cellulose VS, respectively. There are several plausible explanations for this finding.

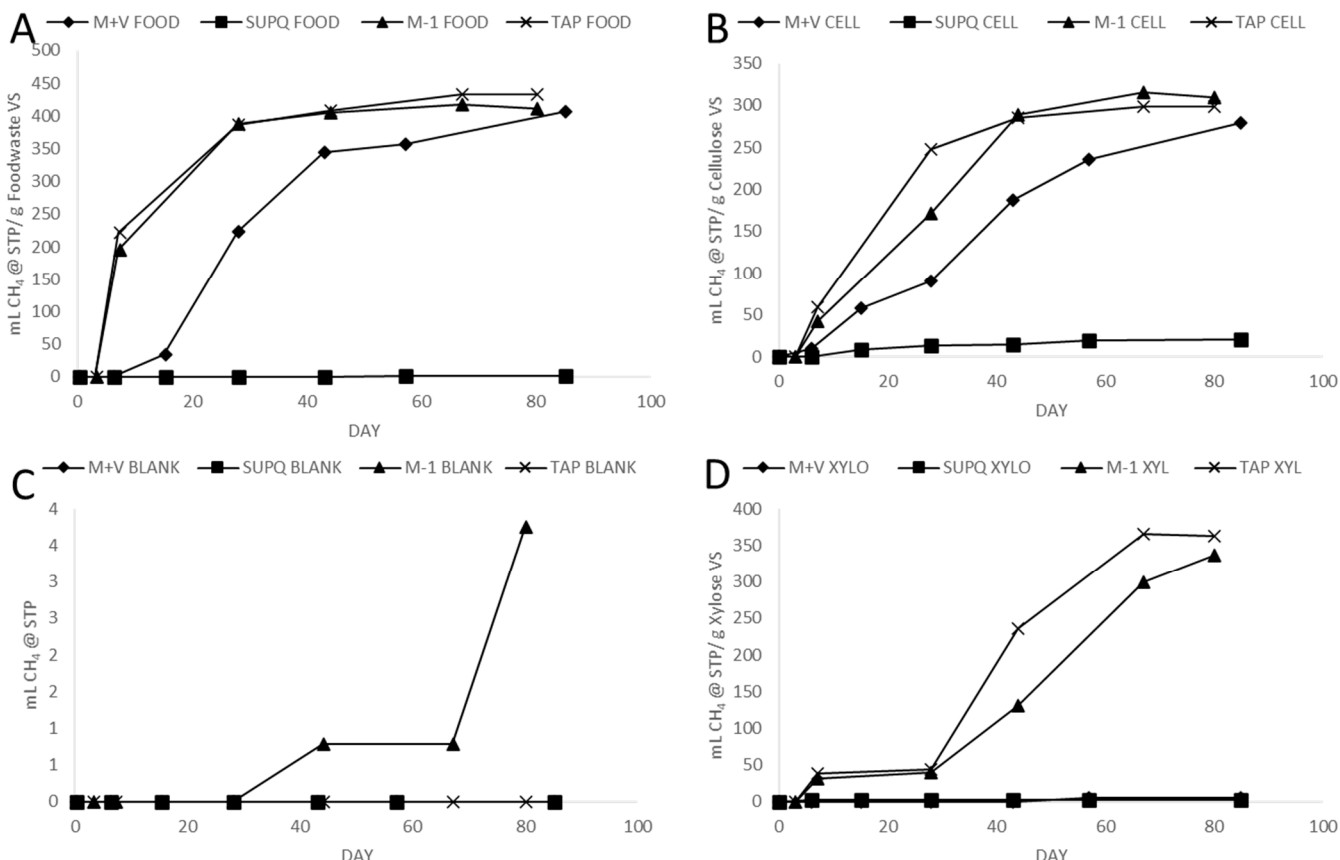

**Figure 3.** BMP results of various growth medias with various substrates. (**A**) shows BMP results with freeze-dried food waste as a substrate with M-1 and M+V media as well as 10% sludge in tap water and SuperQ ultrafiltered water. (**B**) shows the same test with powdered cellulose as a substrate. (**C**) shows blanks and (**D**) shows the BMP results with xylose. All tests are triplicate averages.

**Table 3.** Components of BMP growth medias in this project. All chemicals listed without ** notation are included in the M-1 media. Chemicals with ** are included in only the M+V mixture. Both medias are assembled the same way; M+V incorporates the chemicals noted with ** as a final step.

| Chemical | Concentration (g/L) |
|---|---|
| Resazurin | 0.5 |
| $(NH_4)_2HPO_4$ | 10 |
| $CaCl_2$ | 10 |
| $NH_4Cl$ | 100 |
| $MgCl_2 \cdot 6H_2O$ | 60 |
| KCl | 10 |
| $MnCl_2 \cdot 4H_2O$ | 0.02 |
| $CoCl_2 \cdot 6H_2O$ | 0.017 |
| $H_3BO_3$ | 0.05 |
| $CuCl_2$ | 0.05 |
| $Na_2MoO_4 \cdot 2H_2O$ | 0.05 |
| $ZnCl_2$ | 0.05 |
| $FeCL_2 \cdot 4H_2O$ | 20 |
| $Na_2S \cdot 9H_2O$ | 50 |
| ** Biotin | 0.002 |
| ** Folic acid | 0.002 |
| ** Pyridoxine acid | 0.01 |
| ** Riboflavin | 0.005 |
| ** Thiamine hydrochloride | 0.005 |
| ** Nicotinic acid | 0.005 |
| ** DL-Pantothenic acid | 0.005 |
| ** Cyanocobalamine (Synthetic $B_{12}$) | 0.0001 |
| ** *p*-aminobenzoic acid | 0.005 |
| ** Lipoic acid | 0.005 |
| $K_2HPO_4$ | 88 |
| $NiCl_2 \cdot 6H_2O$ | 0.05 |
| $Na_2SeO_3$ | 0.01 |
| $NaHCO_3$ | 16.8 |
| $KH_2PO_4$ | 69 |
| KI | 1 |
| $NaVO_3 \cdot nH_2O$ | 0.05 |

In Experiment 1, sludge was collected from an active wastewater treatment plant anaerobic digester six days prior to starting the BMPs used to collect data. In Experiment 3, BMPs were assembled with the same sludge after it had been maintained in a laboratory digester for 140 days at a much lower feeding rate than what would be expected in an active AD facility. The laboratory digesters maintained hydraulic retention times of 30–40 days and were maintained with daily feedings of approximately 0.25 g volatile solids (VS)/L reactor volume of food waste. This organic loading rate was intentionally lower than a common rate of 1 g VS/L reactor volume to avoid producing unnecessary volumes of gas, requiring careful management. While the sludge was still active enough to provide results, the amount of remaining organic material and vital nutrients should have decreased over

time, leading to a 93% methane production decrease between M-1 and SuperQ triplicates with cellulose in Figure 3B. A pilot experiment, shown in Figure 4, illustrates how six-day-old sludge can produce strong BMPs in both SuperQ and tap water, as well as both growth medias used in this experiment. Further, important to note is that M+V, while still slightly lower in yield in Figure 4, had a less significant decreasing effect on the ultimate yields with fresh sludge.

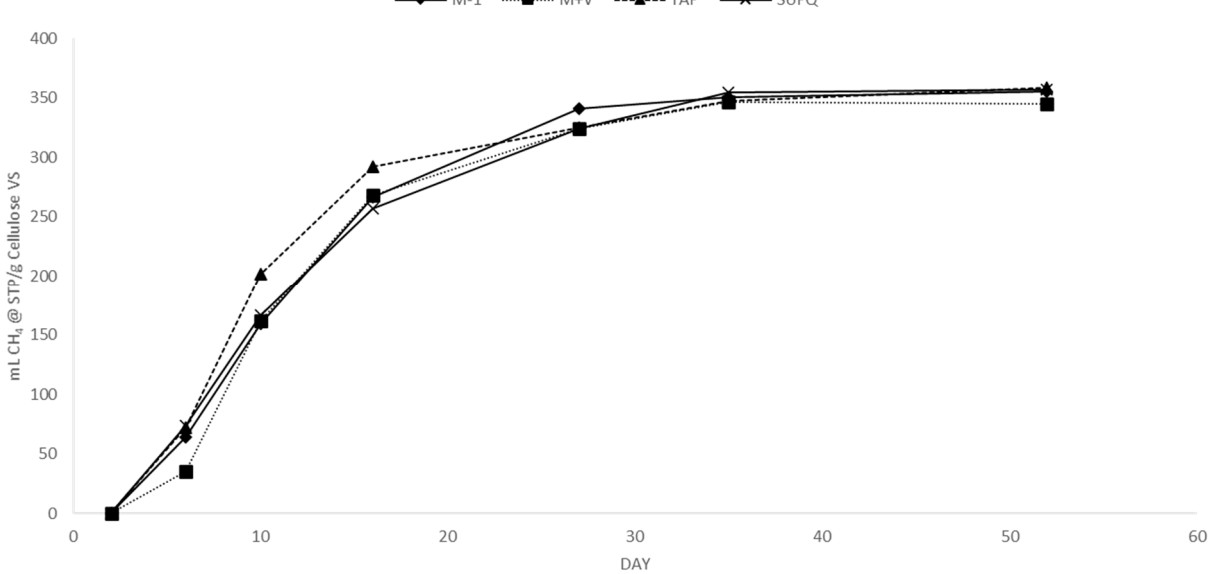

**Figure 4.** Water and media pilot. Mixtures of 10% fresh sludge, collected six days prior to starting BMPs, was added to 90% water or growth media by volume with cellulose substrate in this early pilot experiment.

## 4. Materials and Methods

BMP assays quantify the maximum amount of methane that can be yielded from a biodegradable substrate under idealized anaerobic methanogenic conditions. A key focus in methodologies for BMPs and other experiments is repeatability under similar conditions. While factors such as temperature, sample mass, and sludge volume can be repeated without extensive preparation, the sources and conditions of the methanogen-rich sludge are difficult to replicate among laboratories in different regions. Sludge sourced from wastewater treatment plants, active anaerobic digesters, or other anaerobic environments can all convert biodegradable substrates to methane; however, the rates and efficiencies of each sludge can differ. Much of the research in this project was motivated by the hypothesis that using a small volume of sludge relative to a large volume of pre-made anaerobic growth media mix (10:90 sludge:media by volume) could reduce variation in methane yields among samples, allowing more repeatability and improving the likelihood that sludge will produce the maximum methane yield from substrates. This hypothesis was tested with a variety of factors in a series of experiments described in the following sections.

### 4.1. Anaerobic Sludge Seed

Samples of active anaerobic sludge were collected from the City of Fairfield, Ohio, USA Wastewater Treatment Plant. A continuously stirred (CSTR) anaerobic digestion system treats municipal sewage sludge on site. Active sludge was collected directly from the digester using a sample port built into the system. Samples were collected using five-gallon water-tight Nalgene carboy containers and transported at room temperature for less than one hour before storing in the 37 °C-laboratory incubator.

The sludge sample was off-gassed for 3 days in the incubator, avoiding pressure buildup in the vessel by venting excess gas with a water trap. The off-gassed sludge was

transferred to a 15 L freestanding anaerobic digester with stirring paddles that rotated at 38 RPM for 10 min each hour. Digesters were fed blender-ground freeze-dried cafeteria food waste that had been kept frozen in whole form. Hydraulic retention times of 30–40 days were maintained with daily feedings of approximately 0.25 g volatile solids (VS)/L reactor volume of food waste. The effluent was replaced with equal volumes of deionized (DI) water. Sludge was analyzed, directly prior to use for BMP inoculation, pH and conductivity (Thermo Scientific Orion Probes, Beverly, MA, USA). Dilutions of the sludge were analyzed for chemical oxygen demand and ammonia nitrogen ($NH_3$-N) (HACH Test Kits, Loveland, CO, USA).

### 4.2. Biochemical Methane Potential Assays

The BMP assay bottles were assembled and measured using a methodology largely based on ASTM E 1196-92 and OPPTS 835.3420 standard methods [4,6]. BMP assays were performed in 200 mL glass serum bottles (Wheaton, Millville, NJ, USA) with aluminum crimped rubber septa. A total working volume of 100 mL of homogenized sludge and specified media/water was added under anaerobic conditions, produced by sparging the filled bottles with ultra-high-purity nitrogen gas (AirGas, Cincinnati, OH, USA) for several minutes before capping. BMPs were incubated at 37 °C or 60 °C for the length of the experiment and only removed from the incubators for several minutes to analyze gas composition.

Gas production was quantified in each bottle over a span of 60–95 days for most experiments by measuring the volume displaced via syringe and composition via gas chromatograph. The gas volume was measured in the incubator to avoid volume inconsistencies resulting from temperature changes using a glass syringe (100 mL, Cadence Science, Cranston, RI, USA) and needle (22G) that allowed the pressure in the headspace to equalize with the atmosphere once the septum was pierced. The gas volume was paired with composition data gathered using an Agilent 6890N GC-TCD (Santa Clara, CA, USA) with columns selected to output readings for $H_2$, $O_2$, $N_2$, $CO_2$, and $CH_4$. These values were paired with atmospheric pressure readings in the room at the time of measurement and the volume of the headspace to calculate the amount of each gas produced. These measurements were added cumulatively over time to determine methane potential in different substrates.

According to Pham et al. [11], a successful BMP will reach 80% of the theoretical methane potential, which is usually calculated using a stoichiometric formula such as that suggested by Buswell and Mueller, later updated by Boyle and others [16–18]. Cellulose is commonly used as a control substrate in BMP experiments due to its ease of obtaining and simplified chemical formula, estimated to produce a theoretical BMP of 414 L $CH_4$/kg [19]. Most researchers find the yield of cellulose in BMPs to max out between 350 and 400 mL $CH_4$/g VS in varying conditions and trials [10]. The differences between the theoretical and actual BMP values could be attributed to differences in reactor conditions, substrates, and products containing elements other than carbon, hydrogen, oxygen, nitrogen, and sulfur (CHONS). The presence of indigestible substrates, sometimes referred to as ash, also contributes to differences between the theoretical and observed methane yields [16]. The equations used for theoretical calculations also assume a complete conversion of biomass [17,19]. The methanogens breaking down these substrates need to obtain their own nutrients to survive and cannot convert all of the substrates completely to biogas without metabolizing some substrate for their own needs.

### 4.3. BMP Growth Media

The growth media used for these experiments combines the main constituents of the media defined in the ASTM E1196-92 protocol and a method described by Angelidaki and Sanders in a 2004 review of methods [4,13]. Two different medias were used in this project, labeled "M-1" and "M+V". M-1 was a mixture of chemicals in ultrafiltered (Millipore Super-Q) and autoclaved deionized water. M+V is the same as M-1 with an added mix of

trace vitamins. Table 3 lists the components of each growth media, in which the ** notation indicates constituents only found in the M+V media. The growth media and sludge were added to the serum bottle with approximately 0.2 g VS of the substrate to be digested, including lab-grade reagents in dry, powdered form or freeze-dried food waste collected and frozen for storage. The autoclaved mixture was sparged with ultra-high-purity N2 gas (AirGas, USA) in the serum bottle to flush oxygen out of the headspace and capped with a rubber septum and aluminum crimped cap. The bottles were kept in a 37 °C or 60 °C incubator and stored stationary.

The growth media components selected in Table 3 were chosen after a review of published growth medias, with their formulas listed in Table 4. The review further demonstrated the similarities and differences in chemical selection, notation, and molecular mass used in each growth media. Many mixtures appear to be based on the Owen et al. [1] protocol, which likely based the vitamin mixture on a formula from Wolin, Wolin, and Wolfe [20]. Wolin et al. [20] based their formula on the work of Johns and Barker [21] but added their own mixtures of "vitamins" and "minerals" with no referenced source for how the concentrations of each compound were determined. If there was the reasoning for the selected concentrations of vitamins and minerals used, it was not explained in the publication but was used as the foundation for most BMP protocols developed since. While many of the chemicals have remained the same in growth medias developed over time, the differences in concentrations between some of the authors' mixtures are multiple orders of magnitude. How most of these researchers determined the amount of each chemical used is not explicitly described in their respective works.

**Table 4.** Comparisons of published growth medias. Chemical names with the (") notation are assumed to be either homonyms or close substitutes for the chemical listed in the above table. The authors may be referencing the same compound but used a different naming convention or only had access to the hydrated version or a similar substitute. The (") notation indicates an assumption and is not absolute, as this review is based solely on the information available in each published paper. Masses of each chemical with the (*) or (#) notation indicate that it was categorized by the column's authors as one of two groups of components, indicated below each author in the respective column. Most chemicals included in the M-1 mixture are listed with (#), while the trace vitamins of the M+V chemicals mostly have a (*) notation.

| | Owen et al. (1979) [1] | Wolin et al. (1963) [20] | RAMM by Shelton and Tiedje (1984) [22] | ASTM E1196-92 (ASTM, 1992) [4] | Pagga and Beimborn (1993) [12] | Angelidaki and Sanders (2004) [13] | Pham et al. (2013), Based on VDI (2006) and ISO 11734 (ISO, 1995) [7,13,23] | ISO 11734 (ISO, 1995) [13] |
|---|---|---|---|---|---|---|---|---|
| | S4 (#) | "Minerals" (#) | "Mineral salts" (#) | | | "Trace-metal/selenite" (#) | | |
| | S7 (*) | "Vitamins" (*) | "Trace metals" (*) | "Trace Metals" (*) | "Trace Elements" (*) | "Vitamin mix" (*) | | "Trace Elements" (*) |
| Formula/Chemical Name | Mass (g/L) | Mass (g/L) | Mass (g/L) | Mass (g/L) | Mass (g/L) | Mass (g/L) | Mass (g/L) | Mass (g/L) |
| Resazurin | 1 | | | 0.5 | 0.001 | 0.5 | | 0.001 |
| $(NH_4)_2HPO_4$ | 26.7 | | | 10 | | | | |
| $CaCl_2 \cdot 2H_2O$ | 16.7 (#) | | 0.075 (#) | | 0.075 | 5 | 0.075 | 0.075 |
| $CaCl_2$ (") | | 0.1 (#) | | 10 (*) | | | | |
| $NH_4Cl$ | 26.6 (#) | | 0.53 (#) | 100 | 0.53 | 100 | 0.53 | 0.53 |
| $MgCl_2 \cdot 6H_2O$ | 120 (#) | | 0.1 (#) | 60 (*) | 0.1 | 10 | 0.1 | 0.1 |
| KCl | 86.7 (#) | | | 10 (*) | | | | |
| $MnCl_2 \cdot 4H_2O$ | 1.33 (#) | | 0.0005 (*) | 0.02 (*) | 0.0005 (*) | 0.05 (#) | | 0.05 (*) |
| $CoCl_2 \cdot 6H_2O$ | 2 (#) | | 0.0005 | 0.017 | 0.00001 | 0.05 (#) | | 0.1 (*) |
| $CoCl_2$ (") | | 0.1 (#) | | | | | | |
| $H_3BO_3$ | 0.38 (#) | 0.01 (#) | 0.00005 (*) | 0.05 (*) | 0.00005 (*) | 5 (#) | | 0.005 (*) |
| $CuCl_2 \cdot 2H_2O$ | 0.18 (#) | | | | | 0.038 (#) | | |
| $CuCl_2$ (") | | | 0.00003 (*) | 0.05 (*) | 0.00003 (*) | | | 0.003 (*) |
| $Na_2MoO_4 \cdot 2H_2O$ | 0.17 (#) | | | 0.05 | | | | |
| $Na_2MoO_4$ (") | | 0.01(#) | | | | | | |
| $Na_2Mo_4 \cdot 2H_2O$ (") | | | 0.00001 (*) | | | | | |
| $Na_2MoO_4 \cdot 2H_2O$ (") | | | | 0.05 (*) | 0.00001 (*) | | | 0.001 (*) |
| $ZnCl_2$ | 0.14 (#) | | 0.00005 (*) | 0.05 (*) | 0.00005 (*) | 0.05 (#) | | 0.005 (*) |
| $FeCL_2 \cdot 4H_2O$ | 370 | | 0.02 (#) | 20 (*) | 0.02 | 2 (#) | 0.02 | 0.02 |
| $Na_2S \cdot 9H_2O$ | 500 | | 0.5 | 50 | 0.1 | For reducing to a concentration of 0.25% | 0.1 | 0.1 |

**Table 4.** *Cont.*

| | Owen et al. (1979) [1] | Wolin et al. (1963) [20] | RAMM by Shelton and Tiedje (1984) [22] | ASTM E1196-92 (ASTM, 1992) [4] | Pagga and Beimborn (1993) [12] | Angelidaki and Sanders (2004) [13] | Pham et al. (2013), Based on VDI (2006) and ISO 11734 (ISO, 1995) [7,13,23] | ISO 11734 (ISO, 1995) [13] |
|---|---|---|---|---|---|---|---|---|
| Biotin | 0.002 (*) | 0.002 (*) | | | | 0.002 (*) | | |
| Folic acid | 0.002 (*) | 0.002 (*) | | | | 0.002 (*) | | |
| Pyridoxine hydrochloride | 0.01 (*) | 0.01 (*) | | | | | | |
| Pyridoxine acid (") | | | | | | 0.01 (*) | | |
| Riboflavin | 0.005 (*) | 0.005 (*) | | | | 0.005 (*) | | |
| Thiamin | 0.005 (*) | | | | | | | |
| Thiamine (") | | 0.005 (*) | | | | | | |
| Thiamine hydrochloride (") | | | | | | 0.005 (*) | | |
| Nicotinic acid | 0.005 (*) | 0.005 (*) | | | | 0.005 (*) | | |
| Pantothenic acid | 0.005 (*) | 0.005 (*) | | | | | | |
| DL-Pantothenic acid (") | | | | | | 0.005 (*) | | |
| $B_{12}$ | 0.0001 (*) | 0.0001 (*) | | | | | | |
| Cyanocobalamine (Synthetic $B_{12}$) (") | | | | | | 0.0001 (*) | | |
| *p*-aminobenzoic acid | 0.005 (*) | 0.005 (*) | | | | 0.005 (*) | | |
| Thioctic acid | 0.005 (*) | 0.005 (*) | | | | | | |
| Lipoic acid (") | | | | | | 0.005 (*) | | |
| $N(CH_2CO_2H)_3$ | | 1.5 (#) | | | | | | |
| $MgSO_4$ | | 3 (#) | | | | | | |
| $MnSO_4$ | | 0.5 (#) | | | | | | |
| NaCl | | 1 (#) | | | | 10 | | |
| $FeSO_4$ | | 0.1 (#) | | | | | | |
| $ZnSO_4$ | | 0.1 (#) | | | | | | |
| $CuSO_4$ | | 0.01 (#) | | | | | | |
| $AlK(SO_4)_2$ | | 0.01 (#) | | | | | | |
| $K_2HPO_4 \cdot 3H_2O$ | | | | | | 200 | | |
| $K_2HPO_4$ (") | | | 0.35 | 88 | | | | |
| $Na_2HPO_4 \cdot 12H_2O$ (") | | | | | 1.12 | | | 1.12 |

**Table 4.** *Cont.*

| | Owen et al. (1979) [1] | Wolin et al. (1963) [20] | RAMM by Shelton and Tiedje (1984) [22] | ASTM E1196-92 (ASTM, 1992) [4] | Pagga and Beimborn (1993) [12] | Angelidaki and Sanders (2004) [13] | Pham et al. (2013), Based on VDI (2006) and ISO 11734 (ISO, 1995) [7,13,23] | ISO 11734 (ISO, 1995) [13] |
|---|---|---|---|---|---|---|---|---|
| $(NH_4)_6Mo_7O_{24} \cdot 4H_2O$ | | | | | | 0.05 (#) | 1.12 | |
| $AlCl_3$ | | | | | | 0.05 (#) | | |
| $NiCl_2 \cdot 4H_2O$ | | | | | | | | |
| $NiCl_2 \cdot 6H_2O$ (") | | | 0.00005 (*) | 0.05 (*) | 0.0001 (*) | 0.092 (#) | | 0.01 (*) |
| "EDTA" $C_{10}H_{12}N_2O_8{}^{-4}$ | | | | | | 0.5 (#) | | |
| HCl | | | | | | 1 mL of "concentrated" (#) | | |
| $Na_2SeO_3 \cdot 5H_2O$ | | | | | | 0.1 (#) | | |
| $Na_2SeO_3$ (") | | | 0.00005 (*) | 0.01 (*) | 0.000005 (*) | | | 0.005 (*) |
| $C_3H_8ClNO_2S$ | | | | | | 0.5 | | |
| $NaHCO_3$ | | | 1.2 | 16.8 | | 2.6 | | |
| $KH_2PO_4$ | | | 0.27 | 69 | 0.27 | | | 0.27 |
| KI | | | | 1 (*) | | | 0.27 | |
| $NaVO_3 \cdot nH_2O$ | | | | 0.05 (*) | | | | |

## 5. Conclusions

The universal adoption of a single BMP protocol may never become a reality as the substrates tested with these assays can be as diverse as the sources of anaerobic digesting sludge used to generate methane. We know that many factors can influence the results of BMP assays, including the source and type of anaerobic seed, the dimensions of the experiment vessel, and the methods of measurement and data processing [24]. One step in BMP protocols that have been suggested as optional despite having significant potential impacts is the addition of added nutrients in the form of growth media. This project sought to determine if adding a synthetic growth media, as multiple researchers have in the past, is a valuable and necessary step in the process. Based on the findings presented in this research, several recommendations are suggested.

A mixture of 10% sludge to 90% media or water by volume was determined to be optimal. This avoids the large methane volumes generated by blank controls in 100% and 50% that need to be subtracted to normalize the samples and account for background gas production. While 5% sludge dilutions can help avoid this step, the weaker concentration of sludge takes longer to reach the ultimate potential, while a 10% solution completes the test in a similar timeframe as 100% and 50% BMPs. The standard deviation is also lower and more consistent in 10% bottles, while high-concentration sludge BMPs vary more in the early stages of the assay and 5% bottles show less consistent gas production toward the end of each trial. Selecting the optimal sludge-to-media ratio can help narrow this deviation.

Regarding the necessity of artificial growth media, these findings suggest that fresh, strong, active sludge may not see the benefits of adding these nutrients; however, aged sludge in Experiment 3 showed a clear performance improvement with added media. In most cases, it is unclear whether BMP tests will have sufficient nutrients available from the sludge and substrate or if additional supplements are necessary [25]. The M-1 media showed higher methane production than samples prepared using ultrafiltered water as a growth media. While it is unclear why, the added vitamin mix in M+V media consistently hindered ultimate methane potential in each condition of Experiment 3 relative to the results of the M-1 mixture. The M-1 mixture showed no significant negative impacts on methane production with any age sludge. For these reasons, it is recommended that researchers use BMP protocols with 10% sludge:90% M-1 media volume to reduce variability and avoid lower gas yields.

**Author Contributions:** Conceptualization, G.C. and T.T.; methodology, G.C. and T.T.; validation, G.C. and T.T.; formal analysis, G.C.; investigation, G.C.; resources, G.C. and T.T.; data curation G.C.; writing—original draft preparation, G.C.; writing—review and editing, G.C. and T.T.; visualization, G.C. and T.T.; supervision, T.T.; All authors have read and agreed to the published version of the manuscript.

**Funding:** The funding used for this research was appropriated to US EPA ORD by US Congress.

**Institutional Review Board Statement:** Not applicable.

**Informed Consent Statement:** Not applicable.

**Data Availability Statement:** Additional data may be available from authors upon approved request.

**Conflicts of Interest:** The authors have no conflict of interest or competing interests to declare that are relevant to the content of this article.

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
