# Peer review of "Growth Media Efficacy in Biochemical Methane Potential Assays"

_methane, doi:10.3390/methane2020013_

Round 1

Reviewer 1 Report

This paper, entitled Growth Media Efficacy in Biochemical Methane Potential Assays, is a scholarly work and can increase knowledge on this domain. The authors provide an interesting study, the content is relevant to Methane.

I have some general and specific comments :

- The abstract and keywords are meaningful.

- The manuscript is quite well written but from my point unsufficiently related to exisiting literature, there's many papers dealing with similar topics and contents. Please refer to the several papers, I propose for example to refer to Holliger et al., 2016 Water Science and Technology; Hafner et al., 2020 Water; Ribeiro et al., 2020 Water; Filer et al. , 2019 Water ;  and so many others.

- Please check the whole text, there's no insertions of reference, all the reference didn't appear and there's always the same error message "Error! Reference source not found".

- I disagree with the statement in the abstract "the absence of a universally accepted protocol led to this project...". There's many papers dealing with proposal of harmonized protocols coming from the AD community and especially AD specialist group of IWA, and there's also many norma or guidelines (ISO, VDLUA, ASTM, DIN, ...). Please refer to all these works and studies.

- In the paper Ribeiro et al., 2020 Water, the authors shown that the use of nutrients or not have low effect on BMP ("The sole parameter showing a low effect, higher BMP values (about 9% for the three substrates with limited p-values), was the use or not of the mineral nutrient solution, more than the use of a buffer solution.").

- Please provide error bars for graphs.

- From my point of view, I consider that this paper should be published but  requires many amendments before, especially the state-of-art must be improved and more related to existing literature. The authors have to consider and to discuss their results according to the numerous studies dealing with same topic. The authors have to highlight the originality of their work and to show how this study could provide new knowledge on this topic.

- List of references must be completed with many other studies and papers already published on this topic, I advice the authors to consider some papers such as Hafner, Holliger, Filer and so on.

As it, this paper is not fully acceptable for publication and requires some amendments and additional information. I recommend the following decision: RECONSIDER AFTER MAJOR REVISION.

-

Author Response

Author's Reply to the Review Report (Reviewer 1)

Manuscript ID: methane-2274695

Type: Article

Title: Growth Media Efficacy in Biochemical Methane Potential Assays

Authors: Giles William Chickering *, Thabet M Tolaymat

This paper, entitled Growth Media Efficacy in Biochemical Methane Potential Assays, is a scholarly work and can increase knowledge on this domain. The authors provide an interesting study, the content is relevant to Methane.

The authors appreciate the acknowledgement

I have some general and specific comments :

Each is addressed accordingly

- The abstract and keywords are meaningful.

The authors appreciate the acknowledgement

- The manuscript is quite well written but from my point unsufficiently related to exisiting literature, there's many papers dealing with similar topics and contents. Please refer to the several papers, I propose for example to refer to Holliger et al., 2016 Water Science and Technology; Hafner et al., 2020 Water; Ribeiro et al., 2020 Water; Filer et al. , 2019 Water ;  and so many others.

Most of the literature reviewed in the initial draft targeted papers presenting formulas for growth media mixtures. These additional recommendations are appreciated and were very beneficial to review.

The authors have reviewed the listed recommendations and several others. The recommended papers are cited in the amended text.

- Please check the whole text, there's no insertions of reference, all the reference didn't appear and there's always the same error message "Error! Reference source not found".

The authors used EndNote as reference management software and all these references appear to work in the WordDoc we submitted. In addition to checking once more upon submission, the authors can also include a PDF version to see if the error is coming from the Word Doc itself.

- I disagree with the statement in the abstract "the absence of a universally accepted protocol led to this project...". There's many papers dealing with proposal of harmonized protocols coming from the AD community and especially AD specialist group of IWA, and there's also many norma or guidelines (ISO, VDLUA, ASTM, DIN, ...). Please refer to all these works and studies.

The authors agree the wording in the draft abstract did not convey the implied message. Many of the protocols are based on similar concepts, which is described further in the introduction.

In reference to the listed AD specialist groups and their respective protocols, the point of this entire study is that there are many different BMP assay protocols with different nuances, all of which are based around several main concepts. These groups and some of their specific protocols are already listed and referenced in the introduction section.

- In the paper Ribeiro et al., 2020 Water, the authors shown that the use of nutrients or not have low effect on BMP ("The sole parameter showing a low effect, higher BMP values (about 9% for the three substrates with limited p-values), was the use or not of the mineral nutrient solution, more than the use of a buffer solution.").

What’s actually more significant is that Ribeiro’s study shows similar results to our findings, in which Ribeiro’s undefined mineral nutrient solution actually showed a mild decrease in BMP, similar to our M+V solution. This finding has been referenced in our results and discussion section.

- Please provide error bars for graphs.

Due to the combined layout of our graphs, the addition of error bars makes them too cluttered and the graphs much more difficult to read. The inclusion of the standard deviation charts serves to illustrate the spread among replicates.

- From my point of view, I consider that this paper should be published but  requires many amendments before, especially the state-of-art must be improved and more related to existing literature. The authors have to consider and to discuss their results according to the numerous studies dealing with same topic. The authors have to highlight the originality of their work and to show how this study could provide new knowledge on this topic.

The authors appreciated the recommendation for publishing. References suggested by the reviewer in previous comments have been considered and discussed throughout the revised draft.

Pertaining to this comment and others, this work focuses specifically on the efficacy of growth media as opposed to overall optimization/finding a consensus on a protocol. The suggested references are more holistic and after additional review, mostly eschew the topic of growth media and its impacts. This has been added to the review and discussion.

- List of references must be completed with many other studies and papers already published on this topic, I advice the authors to consider some papers such as Hafner, Holliger, Filer and so on.

The authors appreciated the recommended papers. The reviewed papers are cited in the amended text.

As it, this paper is not fully acceptable for publication and requires some amendments and additional information. I recommend the following decision: RECONSIDER AFTER MAJOR REVISION.

The authors appreciate the feedback and believe the revised draft is much stronger after receiving the reviewer’s comments.

Reviewer 2 Report

1. The significance of the study is unclear.

2. The novelty of this manuscript is unclear. Please show the novelty of this manuscript clearly in the introduction.

3. What is the basis for the preparation M-1 synthetic growth media?

4. There are some formatting errors. Such as lines 143,144. Please check the whole manuscript.

5. The quality of the figures should be improved.

6. The difference value between figure 1A and figure 1B should be provided.

7. The experimental result should be compared with other literature.

8. More in-depth discussion is needed in 4. Results and Discussion.

9. Conclusions based on limited experimental data are not reliable.

Author Response

Author's Reply to the Review Report (Reviewer 2)

Manuscript ID: methane-2274695

Type: Article

Title: Growth Media Efficacy in Biochemical Methane Potential Assays

Authors: Giles William Chickering * , Thabet M Tolaymat

1. The significance of the study is unclear.

The authors have amended the abstract and introduction with suggestions from reviewers to improve the clarity of the significance of this study.

2. The novelty of this manuscript is unclear. Please show the novelty of this manuscript clearly in the introduction.

The authors have amended the abstract and introduction with suggestions from reviewers to improve the clarity of the novelty of this study as well. The final sentence of the abstract is now a direct statement of the novelty/takeaway from this paper.

3. What is the basis for the preparation M-1 synthetic growth media?

The “BMP Growth Media” section of the Materials and Methods section gives both a concise literature review and explanation for how this synthetic growth media was determined and the history of growth medias in past BMP studies.

4. There are some formatting errors. Such as lines 143,144. Please check the whole manuscript.

A final formatting audit was performed prior to submitting the revised draft.

5. The quality of the figures should be improved.

The figures are intentionally minimal to avoid overburdening the reader and provide a concise depiction of our summarized data. If the journal needs something specific in terms or formatting, the authors will swiftly adapt the figures to any specified requirements.

6. The difference value between figure 1A and figure 1B should be provided.

The caption explains the difference but this comment led the authors to discover an issue with the Y-axis label that has now been fixed in both Figures 1 and 2.

7. The experimental result should be compared with other literature.

The authors agree and the recommended papers by another reviewer have been added for context and comparison throughout this revision.

8. More in-depth discussion is needed in 4. Results and Discussion.

The authors opted to add more to the conclusion as the results and discussion are getting extensive in length and the final section needed more strength.

9. Conclusions based on limited experimental data are not reliable.

The authors agree and the language has been hedged to reflect this. Additional comparisons to published works were also added to improve this section.

Round 2

Reviewer 1 Report

The authors provide a revised version of their manuscript taking into account all the comments and requests of amendments. They provide also detailed and justified answers to the comments. I agree with all the comments and amendments. I recommend the following decision: ACCEPT IN PRESENT FORM.

Author Response

Thank you for agreeing to accept in present form 

Reviewer 2 Report

The conclusion should be more concise.

Author Response

The conclusion was shortened as recommended and more relevant information was moved and added to the results and discussion. 

A full final check of the paper was also performed to clean up any final wording and details. 

Thank you for this recommendation